## ·ᐧ·PLOS | ONE

# Toll-like receptor 7-adapter complex modulates interferon-α production in HIV-stimulated plasmacytoid dendritic cells

**Andy A. Patamawenu[1◉], Nathaniel E. Wright[1◉], Tulley Shofner[1], Sean Evans[1], Maura M. Manion[1], Nicole Doria-Rose[1], Stephen A. Migueles[1], Daniel Mendoza[1], Bennett Peterson[1], Christopher Wilhelm[1], Julia Rood[1], Amy Berkley[1], Nancy A. Cogliano[1], C. Jason Liang[2], Kiki Tesselaar[3], Frank Miedema[3], Julian Bess, Jr.[4], Jeffrey Lifson[4], Mark Connors[1]\***

1 HIV-Specific Immunity Section of the Laboratory of Immunoregulation, National Institute of Allergy and Infectious Diseases, National Institutes of Health, Bethesda, Maryland, United States of America, 2 Biostatistics Research Branch, National Institute of Allergy and Infectious Diseases, National Institutes of Health, Bethesda, Maryland, United States of America, 3 Department of Immunology, University Medical Center Utrecht, Utrecht, Netherlands, 4 AIDS and Cancer Virus Program, Frederick National Laboratory, Frederick, Maryland, United States of America

◉ These authors contributed equally to this work.
\* mconnors@niaid.nih.gov

**Data Availability Statement:** All relevant data are within the manuscript and at https://doi.org/10.6084/m9.figshare.7827971 (XLSX).

## Abstract

Plasmacytoid dendritic cells (PDCs) and their production of interferon-alpha (IFN-α) are believed to play an important role in human immunodeficiency virus, type I (HIV-1) pathogenesis. PDCs produce IFN-α and other proinflammatory cytokines through stimulation of Toll-like receptor 7 (TLR7) and TLR9 present in endosomal compartments. TLR7 recognizes single-stranded viral RNA, while TLR9 recognizes unmethylated DNA. In this study, we examined the mechanisms that may underlie variations in IFN-α production in response to HIV, and the impact of these variations on HIV pathogenesis. In four distinct cohorts, we examined PDC production of IFN-α upon stimulation with inactivated HIV-1 particles and unmethylated DNA. The signaling cascade of TLR7 bifurcates at the myeloid differentiation protein 88 (MyD88) adaptor protein to induce expression of either IFN-α or TNF-α. To determine whether variations in IFN-α production are modulated at the level of the receptor complex or downstream of it, we correlated production of IFN-α and TNF-α following stimulation of TLR7 or TLR9 receptors. Flow cytometry detection of intracellular cytokines showed strong, direct correlations between IFN-α and TNF-α expression in all four cohorts, suggesting that variations in IFN-α production are not due to variations downstream of the receptor complex. We then investigated the events upstream of TLR binding by using lipid-like vesicles to deliver TLR ligands directly to the TLR receptors, bypassing the need for CD4 binding and endocytosis. Similar tight correlations were found in IFN-α and TNF-α production in response to the TLR ligands. Taken together, these results strongly suggest that differences in IFN-α production depend on the regulatory processes at the level of the TLR7 receptor complex. Additionally, we found no association between IFN-α production before HIV infection and disease progression.

**Funding:** This work was supported by the Intramural Research Programs of the National Institute of Allergy and Infectious Diseases (NIAID), NIH to MC.

**Competing interests:** The authors have declared that no competing interests exist.

## Introduction

One of the many specialized cell types that recognize foreign antigens shortly after infection, plasmacytoid dendritic cells (PDCs) play an important role in the innate immune system's early detection of viral pathogens. PDCs are a relatively rare subset of circulating dendritic cells (DCs) that comprise approximately 0.4% of circulating cells. They are distinguished phenotypically by the expression of the surface markers CD123[+], BDCA-2 (CD303[+]), and BDCA-4 (CD304[+]) as well as the absence of CD11 and CD14, which differentiates them from other DC subsets and monocytes [1, 2]. PDCs represent the major source of interferon-α (IFN-α) production upon recognition of a pathogen [2–4]. IFN-α is a critical, pleiotropic type I interferon that is comprised of 13 subtypes and mediates a wide range of effects including inhibition of replication of numerous viruses including HIV-1 [5–7]. PDCs respond to HIV antigens within hours of contact and mature into professional antigen-presenting DCs within weeks. During viral infections they may be activated through their cell membrane-bound, cytosolic, and endosomal receptors. PDCs express toll-like receptors (TLRs) that can directly recognize a wide range of pathogen associated molecular patterns (PAMPs) [7–9]. TLR7 and TLR9, expressed in PDC endosomes, specifically recognize viral RNA and unmethylated bacterial DNA respectively, while they do not respond to endogenous cellular RNA or DNA [6]. Both TLRs depend on the adaptor protein myeloid differentiation primary-response protein 88 (MyD88) for signaling and PDC activation [9].

In HIV infection, PDCs have been the focus of intense investigation because of their role in the innate immune response against the virus and their potential role in pathogenesis [10, 11]. PDCs bind HIV virions through surface expression of the CD4 receptor that is required for viral entry [11]. When HIV virions enter the PDC endosome, viral RNA binds to TLR7 causing bifurcated signaling downstream through MyD88. These two major pathways below MyD88 are the interferon regulatory factor 7 (IRF7) pathway for type I IFN (mainly IFN-α and IFN-β) production and the NF-κB pathway for TNF-α and other proinflammatory cytokine production [9].

The mechanisms by which HIV-1 activates PDCs remain incompletely understood. In a prior study, we observed wide variations in IFN-α production by PDCs in response to HIV virions [12]. This result is striking given that polymorphisms for TLR7 are not common and should thus provide a similar stimulus across patients. We investigated whether production of different levels of either IFN-α through the IRF7 pathway or TNF-α through the NF-κB pathway in HIV-infected subjects may be caused by variations in PDC activation via the TLR7/MyD88 receptor adaptor complex. In this study, we examined IFN-α production by PDCs in cross-sectional samples from three patient groups: uninfected donors, long-term nonprogressors or elite controllers (LTNP/ECs), and HIV-1-infected progressors. We also examined the potential correlations between variations in IFN-α production in response to inactivated virions and progression of disease. Because HIV viremia can stimulate IFN-α production in vivo resulting in diminished IFN-α production ex vivo, we used samples from participants in the Amsterdam Cohort Studies (HIV-uninfected subjects with high-risk behavior for HIV acquisition) to study this question. We observed that the variations of IFN-α production by PDCs among participants were tightly correlated with production of TNF-α and were independent of viral entry, suggesting this variation is mediated at the level of the TLR7-MyD88 complex. Additionally, we observed that variations in IFN-α production before HIV infection were not directly associated with disease progression.

## Materials and methods

### Study subjects

Four different cohorts were used in this study: (1) twenty healthy, uninfected donors, (2) twenty-three subjects from the Amsterdam Cohort Studies (individuals with IV drug abuse

who provided samples before HIV infection [13]), (3) eleven LTNP/ECs who were defined as HIV-1-infected patients with no prior history of opportunistic diseases, stable CD4 T cell counts, and HIV RNA levels of <50 copies/ml plasma in the absence of anti-retroviral therapy, (4) nine progressors defined as HIV-1-infected patients with a progressive decline in CD4 T cell counts and/or current or previously documented poor restriction of virus replication (HIV RNA levels >1000 copies/ml) prior to the initiation of antiretroviral therapy. At the time of measurement, all progressor patients were on antiretroviral therapy and were stably suppressed to <50 copies/ml plasma. Participant samples were collected from HIV-1-infected LTNP/ECs and progressors from 2001–2008 (Table 1) and participant samples were collected

**Table 1. Characteristics of HIV-infected LTNP/ECs and progressors.**

| Patient Number | Date of Sample | Gender | Diagnosis Year | Risk Factor | Race/ Ethnicity | CD4 T cell counts (cells/ ml) | CD4 T cell counts (%) | Viral Load (copies/ ml) |
|---|---|---|---|---|---|---|---|---|
| **LTNP/ECs** | | | | | | | | |
| 1 | 6/13/05 | M | 1985 | MSM | Caucasian/ Non-Hispanic | 1060 | 41 | <50 |
| 2 | 8/9/04 | M | 1995 | MSM | Caucasian/ Non-Hispanic | 1140 | 33 | <50 |
| 3 | 9/6/01 | F | 1998 | Heterosexual | AA | 1616 | 48 | <50 |
| 4 | 1/30/08 | M | 1996 | IVDU, Heterosexual | AA | 1239 | 44 | <50 |
| 5 | 3/6/07 | F | 1989 | Heterosexual | AA | 601 | 48 | <50 |
| 6 | 9/10/07 | F | 1993 | Heterosexual | AA | 1033 | 60 | <50 |
| 7 | 6/16/06 | F | 1992 | Heterosexual | Caucasian/ Non-Hispanic | 1468 | 49 | <50 |
| 8 | 8/29/07 | M | 1996 | MSM | Caucasian/ Non-Hispanic | 1453 | 43 | <50 |
| 9 | 2/5/08 | M | 1991 | MSM | Caucasian/ Non-Hispanic | 883 | 41 | <50 |
| 10 | 1/3/08 | M | 1987 | MSM | Caucasian/ Non-Hispanic | 655 | 35 | <50 |
| 11 | 12/10/07 | M | 1984 | MSM | Caucasian/ Non-Hispanic | 776 | 36 | <50 |
| **Progressors** | | | | | | | | |
| 1 | 3/13/08 | M | 1997 | Heterosexual | Caucasian/ Non-Hispanic | 567 | 37 | <50[a] |
| 2 | 2/5/08 | F | 1992 | Perinatal | Caucasian/ Non-Hispanic | 850 | 42 | <50 |
| 3 | 3/11/08 | M | 1996 | MSM | Caucasian/ Hispanic | 506 | 19 | <50 |
| 4 | 11/29/05 | M | 2001 | MSM | Caucasian/ Hispanic | 500 | 32 | <50 |
| 5 | 4/8/08 | M | 1994 | MSM | AA | 513 | 27 | <50 |
| 6 | 2/19/08 | M | 1987 | MSM | Caucasian/ Non-Hispanic | 145 | 11 | <50 |
| 7 | 2/25/08 | F | 1986 | IVDU | Caucasian/ Non-Hispanic | 698 | 39 | <50 |
| 8 | 3/11/08 | M | 1986 | MSM | Caucasian/ Non-Hispanic | 276 | 16 | <50 |
| 9 | 2/27/08 | M | 1987 | MSM | Caucasian/ Non-Hispanic | 695 | 35 | <50 |

MSM = Men who have Sex with Men; IVDU = Intravenous Drug User; AA = African American.
[a]Progressor patients were on antiretroviral therapy and were stably suppressed to <50 copies/ml plasma.

**Table 2. Characteristics of HIV-infected subjects from the Amsterdam Cohort studies.**

| Patient Number | Date of Sample | Date of Infection | Viral Load Year 1 (copies/ ml) | CD4+ T Cell Counts Pre-Infection (cells/ml) | CD4+ T Cell Counts Year 1 (cells/ml) | CD4+ T Cell Counts Year 5 (cells/ml) | Change in CD4+ T Cells Years 1–5 | Percent Change in CD4+ T Cells Years 1–5 |
|---|---|---|---|---|---|---|---|---|
| 1 | 9/15/86 | 9/4/91 | 56000 | 820 | 520 | 340 | -180 | 0.35 |
| 2 | 12/3/85 | 4/4/90 | 12000 | 500 | 230 | 10 | 220 | 0.96 |
| 3 | 8/12/86 | 5/11/90 | 14000 | 360 | 80 | n.a. | n.a. | n.a. |
| 4 | 6/11/85 | 8/15/90 | 13000 | n.a. | 1220 | 490 | -730 | 0.60 |
| 5 | 12/19/85 | 8/6/87 | 18000 | 660 | 270 | 200 | -70 | 0.26 |
| 6 | 9/1/86 | 5/15/90 | 36000 | n.a. | 820 | 380 | -440 | 0.54 |
| 7 | 5/28/86 | 11/6/90 | 1000 | 1740 | 1690 | 990 | -700 | 0.41 |
| 8 | 8/26/86 | 8/19/92 | 180000 | n.a. | 620 | 460 | -160 | 0.26 |
| 9 | 9/4/86 | 12/5/89 | 16000 | n.a. | 600 | 300 | -300 | 0.50 |
| 10 | 6/12/86 | 11/2/90 | 1000 | 880 | 950 | 350 | -600 | 0.63 |
| 11 | 10/29/86 | 12/6/90 | 1000 | 1310 | 1090 | 840 | -250 | 0.23 |
| 12 | 9/9/86 | 5/17/94 | 61000 | 580 | 340 | 290 | -50 | 0.15 |
| 13 | 11/4/86 | 4/10/90 | 220000 | n.a. | 540 | 60 | -480 | 0.89 |
| 14 | 3/17/87 | 9/19/91 | 1000 | 590 | 550 | 200 | -350 | 0.64 |
| 15 | 3/4/86 | 2/19/90 | 33000 | 670 | 370 | 480 | 110 | -0.30 |
| 16 | 7/4/86 | 1/16/90 | 16000 | 900 | 460 | 130 | -330 | 0.72 |
| 17 | 1/28/86 | 3/11/95 | 7300 | 760 | 490 | 330 | -160 | 0.33 |
| 18 | 2/12/86 | 8/30/88 | 48000 | 660 | 490 | 310 | -180 | 0.37 |
| 19 | 12/4/86 | 10/10/90 | 17000 | 930 | 750 | 560 | -190 | 0.25 |
| 20 | 3/19/86 | 3/22/95 | 1000 | 1040 | 1120 | 640 | -480 | 0.43 |
| 21 | 4/15/86 | 10/6/87 | 1000 | n.a. | 530 | 600 | 70 | -0.13 |
| 22 | 3/5/86 | 12/2/87 | 160000 | 790 | 400 | 260 | -140 | 0.35 |
| 23 | 5/11/87 | 8/10/87 | 17000 | n.a. | 290 | 30 | -260 | 0.90 |

n.a. = data not available

from participants within the Amsterdam cohort from 1984–2001 for this study (Table 2). Details of some of the LTNP/ECs and progressors have been published previously [14]. All samples were obtained under a protocol that was reviewed and approved by the NIAID Investigational Review Board (IRB) prior to beginning the study. All donors signed NIAID IRB-approved informed consent documents. PBMCs were obtained by leukapheresis as reported previously [12].

## PDC stimulation

In order to measure the downstream signaling of TLR7- and TLR9-MyD88 complexes, four million PBMCs from each subject were stimulated with 250 ng of Aldrithiol-2 (AT-2) inactivated HIV particles (HIV-1$_{ADA}$, a TLR7 agonist), 500 ng of CpG A (a TLR9 agonist), or 250 ng of control microvesicles from SUPT1 cells (Biological Products Core, AIDS and Cancer Virus Program, Frederick National Laboratory, Frederick, MD) in 1 ml of 0.5% human serum (HAB, Gem Cell Gemini Bio-Products, Sacramento, CA) RPMI medium and incubated at 37°C in 5% $CO_2$. After 4 hours of incubation, 1 mg of Brefeldin A (Sigma Aldrich, St. Louis, MO) was added to inhibit cytokine secretion. After 4 additional hours of incubation, reactions were neutralized by adding 100 μl of 20 μM EDTA, pH 8.0. PBMCs were stained with Live/

Dead fixable violet dead cell stain kits (Invitrogen, Carlsbad, CA) and fixed with Cytofix/Cytoperm buffer (BD Biosciences, San Jose, CA) [12].

## RNA41 DOTAP stimulation

Measuring upstream signaling of TLR7- and TLR9-MyD88 complexes by bypassing CD4 required entry was similar to the downstream signaling method above. However, the TLR7 agonist HIV-1$_{ADA}$ was replaced with RNA40 (HIV-1 U5 nucleotide sequences) or RNA41 (a negative control similar to RNA40 in which U was changed to T).

Four million PBMCs from LTNP/ECs and progressors were stimulated in 1 ml of 10% HAB medium containing 1 μg of RNA40 and 6.25 μl DOTAP (Invitrogen). Two negative controls were prepared: (1) 1 μg of RNA40, 1 μg RNase A, and 6.25 μl DOTAP; and (2) 1 μg of RNA41 and 6.25 μl DOTAP.

## Intracellular staining of IFN-α and TNF-α in PDCs

Fixed PBMCs were stained with the following monoclonal antibodies (BD Biosciences): PE-Cy5-anti-CD123 (cat. 551065, RRID: AB_394029), APC-Cy7-anti-HLA-DR (cat. 335796, RRID: AB_399974), FITC-anti-Lineage-1 (cat. 340546, RRID: AB_400053), APC-anti-TNF-α (cat. 554514, RRID: AB_398566), PE-anti-IFN-α (cat. 560097, RRID: AB_1645511) for 30 min and analyzed by flow cytometry on a FACS Aria. Between 1 million to 1.5 million PBMCs were collected. All data was analyzed using FlowJo software (TreeStar, Ashland, OR). PDCs were identified using the following gating strategy: (lineage CD3$^-$ CD8$^-$ CD16$^-$ CD19$^-$ CD56$^-$), HLADR$^+$, CD14$^-$, CD11c$^-$, CD123$^+$ cells (S1 Fig).

## Statistics

Differences in IFN-α production between the four cohorts were analyzed using the Wilcoxon signed-rank test. Correlations with IFN-α and TNF-α production were done by the Spearman rank correlation.

## Results

### PDC production of IFN-α and TNF-α in response to TLR7 and TLR9 agonists

To measure PDC function on a per-cell basis, we employed a flow cytometric intracellular staining technique that we previously developed to measure IFN-α production by PDCs (S1 Fig) [12]. PDC numbers can vary widely between patients and disease states and therefore can affect assays of PDC-produced cytokines measured in culture supernatants. This flow cytometric method potentially permits measurements of production of multiple cytokines within the same cell and avoids the potential impact of PDC frequency [11, 12, 15].

Using this method, we measured IFN-α production in PDCs from 20 HIV-uninfected controls from an NIAID cohort, 23 from the Amsterdam Cohort Studies who provided samples before HIV infection, 11 LTNP/ECs that maintain <50 copies of HIV RNA without antiretroviral therapy, and 9 HIV-infected progressors who were on antiretroviral therapy at the time of measurement (Table 1). Consistent with our prior results, we observed wide variations in the fraction of PDCs that stained for IFN-α following stimulation with AT-2 inactivated HIV-1$_{ADA}$ ex vivo (Fig 1). A greater fraction of PDCs from the uninfected NIAID cohort produced IFN-α in response to HIV-1$_{ADA}$ (median = 4.59%, range = 0 to 67%) compared to the Amsterdam Cohort pre-infection samples (median = 0.55%, range = 0 to 14.57%), LTNP/ECs (median = 1.28%, range = 0 to 4.94%), or progressors (median = 0.55%, range = 0 to 2.59%).

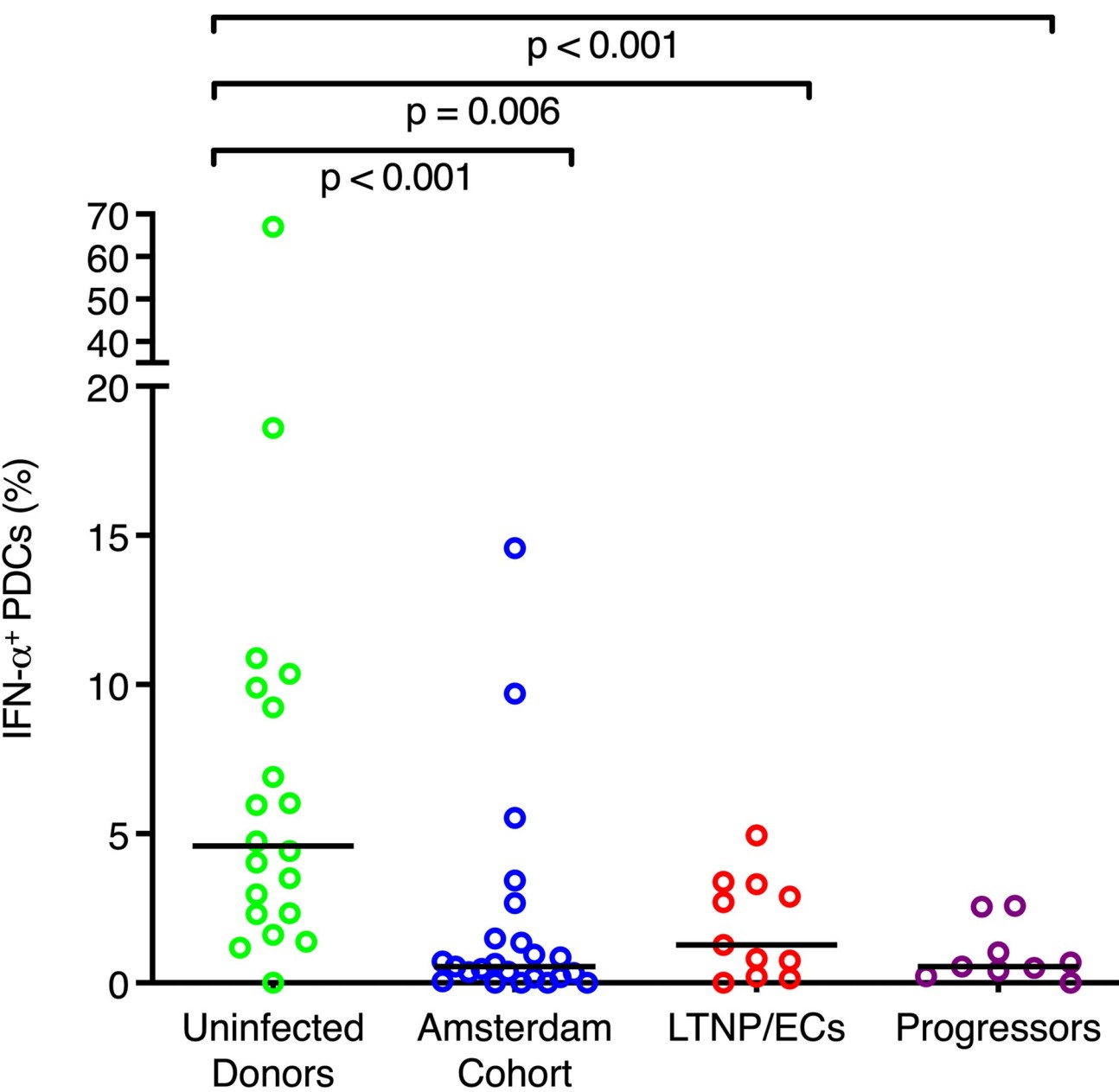

**Fig 1. IFN-α production by PDCs stimulated with HIV-1$_{ADA}$ in four cohorts.** Percentage of PDCs from four different cohorts that stained positive for IFN-α after stimulation with HIV-1$_{ADA}$: uninfected donors, donors from the Amsterdam Cohort Studies sampled prior to HIV-1 infection, LTNP/ECs, and HIV-infected progressors. Horizontal lines indicate median values. P values were calculated with the Wilcoxon signed-rank test. Only p values for comparisons between uninfected donors and the other cohorts are shown.

Although the median fraction of PDCs producing IFN-α was greater in LTNP/ECs than the Amsterdam Cohort or progressors, this difference did not achieve statistical significance. Diminished production of IFN-α by progressors compared to uninfected controls is potentially consistent with prior results suggesting that HIV viremia causes a decrease in the ability of PDCs to respond by producing IFN-α ex vivo [12, 15]. However, it is unclear why the NIAID cohort uninfected donors have higher levels of IFN-α production compared to the HIV-uninfected Amsterdam cohort or LTNP/ECs.

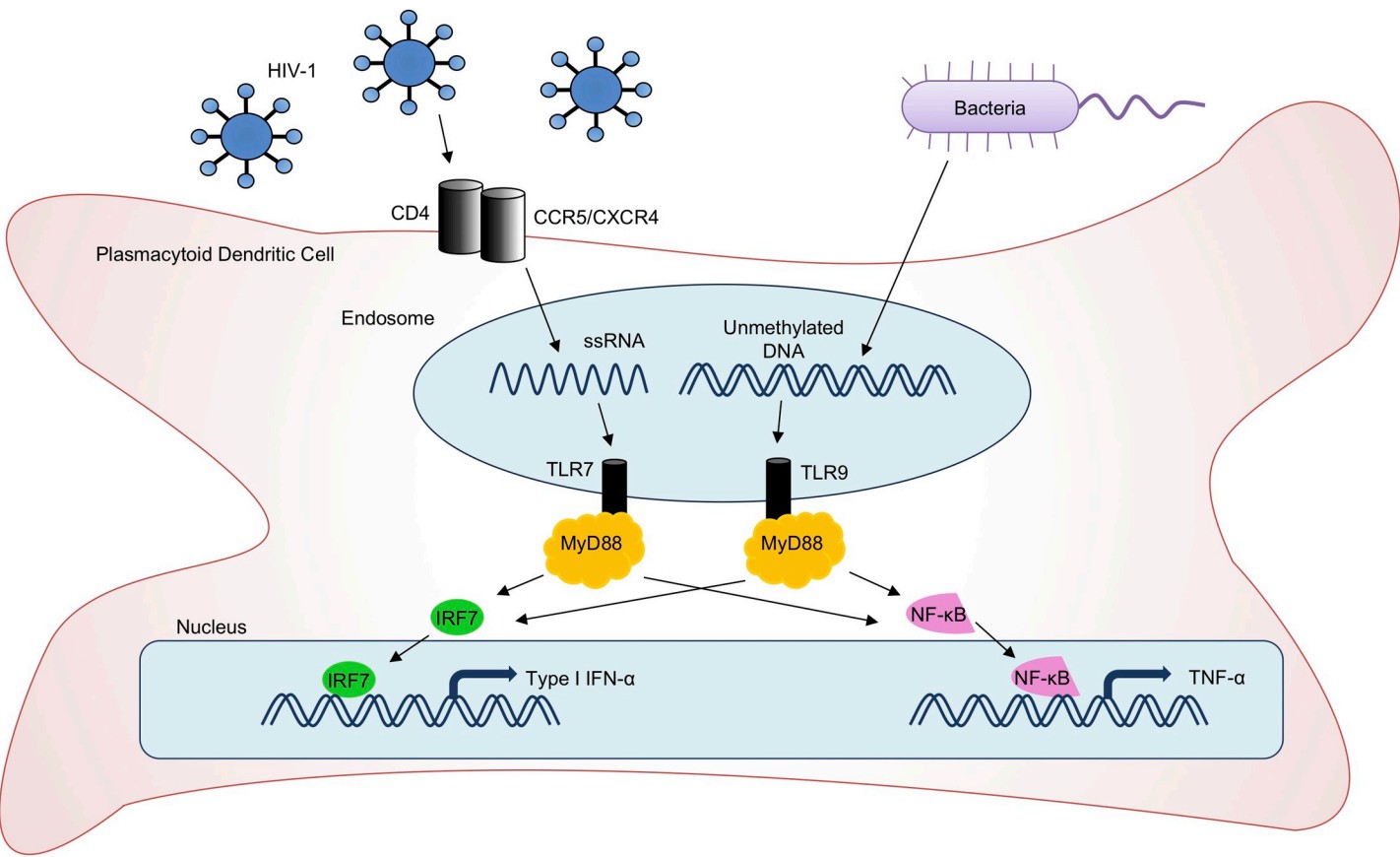

**Fig 2. Signaling pathways of TLR7 and TLR9.** Upon stimulation with viral RNA or unmethylated bacterial DNA, TLRs transduce signals that diverge below MyD88 to cause transcription of IFN-α or TNF-α.

### Differences in IFN-α production in HIV-1$_{ADA}$ stimulated PDCs are not caused by the signaling cascades downstream of the TLR7-MyD88 or TLR9-MyD88 receptor complexes

To understand the potential mechanisms of variation in IFN-α production, we measured both IFN-α and TNF-α production by PDCs in response to stimulation by the TLR7 agonist HIV-1$_{ADA}$ and the TLR9 agonist CpG A to examine the relative expression of IFN-α or TNF-α and to understand whether variations in IFN-α are caused by changes downstream of the TLR7- or TLR9-MyD88 complexes. After the signal is transduced through the MyD88 molecule, IFN-α is regulated through the IRF7 pathway and TNF-α is regulated through the NF-κB pathway (Fig 2). Thus, changes in the IRF7 pathway that might lead to variation in IFN-α would not be expected to affect TNF-α production. However, if expression levels of TNF-α and IFN-α are correlated, it suggests that the observed variations in IFN-α expression between donors may be mediated at or above MyD88, which is shared by both pathways [16, 17].

In response to TLR7 stimulation by HIV-1$_{ADA}$, production of IFN-α and TNF-α by PDCs was tightly correlated in all four cohorts (uninfected donors: $r_s = 0.77$, $p < 0.001$(Fig 3A); Amsterdam cohort: $r_s = 0.92$, $p < 0.001$ (Fig 3B); LTNP/ECs: $r_s = 0.61$, $p = 0.052$ (Fig 3C); progressors: $r_s = 0.97$, $p < 0.001$ (Fig 3D)). Similarly, there were strong correlations between the levels of IFN-α and TNF-α production after TLR9 stimulation with CpG A in the four cohorts (uninfected donors: $r_s = 0.72$, $p < 0.001$(Fig 4A); Amsterdam cohort: $r_s = 0.89$, $p < 0.001$ (Fig

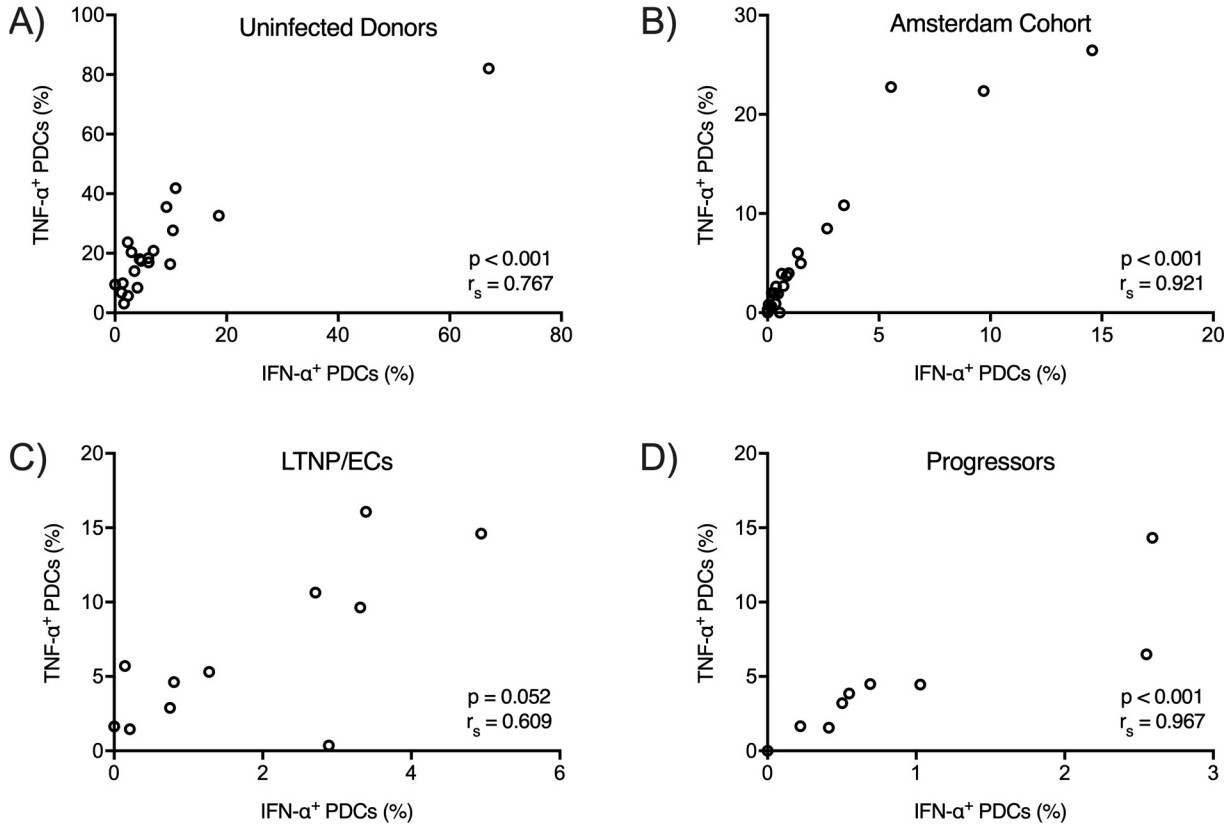

**Fig 3. IFN-α and TNF-α production correlates in PDCs following TLR7 receptor ligation by HIV-1$_{ADA}$.** PDC TLR7 receptors were stimulated by HIV-1$_{ADA}$, and cytokine secretion was inhibited after 4 hours. Anti-IFN-α and anti-TNF-α antibodies were used to measure intracellular IFN-α and TNF-α production by flow cytometry. P and r$_s$ values were calculated by the Spearman rank correlation.

4B); LTNP/ECs: r$_s$ = 0.91, p < 0.001 (Fig 4C); progressors: r$_s$ = 0.83, p = 0.008 (Fig 4D)). In addition, only weak correlations between TLR7- and TLR9-induced IFN-α were observed for uninfected donors (S2A Fig) and the Amsterdam cohort (S2B Fig), which might suggest that variations in IFN-α production were intrinsic to the IRF7 pathway (S2 Fig). There was no similar correlation observed for LTNP/ECs (S2C Fig) or progressors (S2D Fig). Taken together, these results suggested that variations in IFN-α production do not result from signaling differences downstream of the TLR7-MyD88 and TLR9-MyD88 complexes. There may be factors upstream or within the complexes themselves that result in differences in IFN-α production by PDCs.

## Differences in IFN-α production in HIV-1$_{ADA}$ stimulated PDCs were not caused by HIV-1 entry

It remained possible that variations in IFN-α production were mediated by variations in CD4 binding, viral particle entry, or uncoating in the endosome. To test this hypothesis, we stimulated PDCs with RNA40, a short stretch of GU-rich single-stranded HIV RNA derived from the 5' long terminal repeat that potently stimulates PDCs [18]. RNA40 was used in the presence of DOTAP (a cationic lipid that facilitates nucleic acid entry into endosomes) with or without RNase A treatment. If variation was mediated at the level of binding, entry, or uncoating, we predicted that DOTAP should remove this variation. We observed tight correlations between IFN-α and TNF-α production by PDCs from stimulation with RNA40-DOTAP in

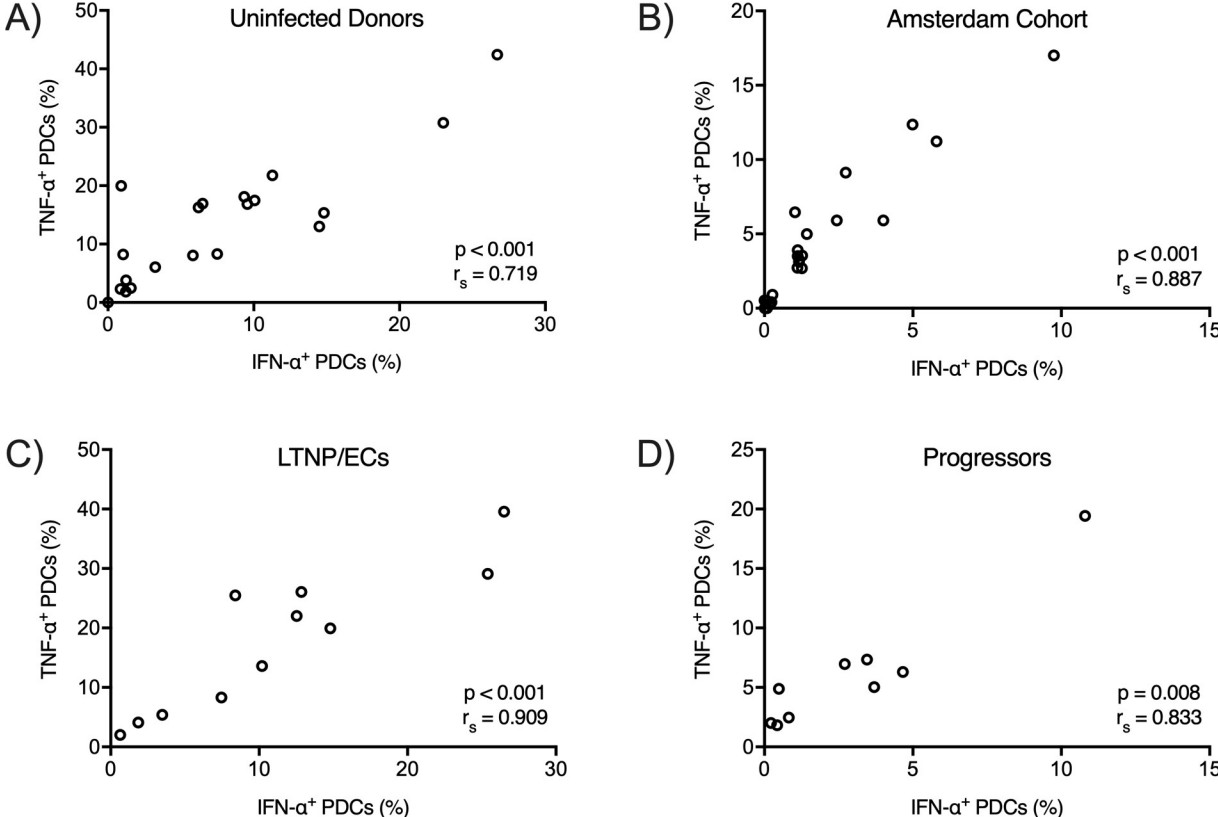

**Fig 4. IFN-α and TNF-α production correlates in PDCs following TLR9 receptor ligation by CpG A.** PDC TLR9 receptors were stimulated by CpG A, and cytokine secretion was inhibited after 4 hours. Anti-IFN-α and anti-TNF-α antibodies were used to measure intracellular IFN-α and TNF-α production by flow cytometry. P and $r_s$ values were calculated by the Spearman rank correlation.

both LTNP/ECs and progressors ($r_s$ = 0.90, p < 0.001 (Fig 5A); $r_s$ = 0.93, p < 0.001 respectively (Fig 5B)), consistent with the results observed when viral particles were used to stimulate PDCs. This result suggests that differences in IFN-α production in different cohorts were not due to variations in viral entry through CD4 binding. Rather, the results are most consistent with variations in IFN-α production modulated at the level of the TLR7-MyD88 complex.

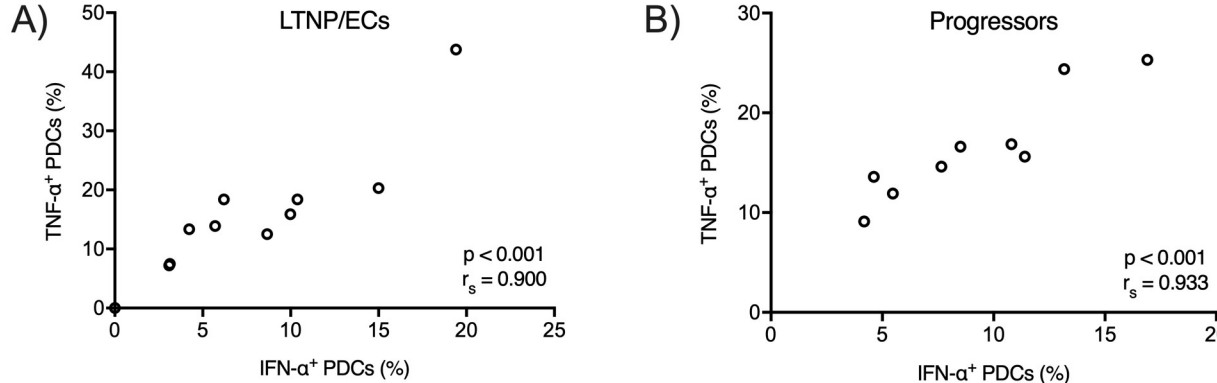

**Fig 5. IFN-α and TNF-α production correlates in PDCs following RNA40-DOTAP stimulation that bypassed CD4-mediated entry.** PDC TLR7 receptors were stimulated by RNA40-DOTAP in order to bypass viral binding, entry, and uncoating. P and $r_s$ values were calculated by the Spearman rank correlation.

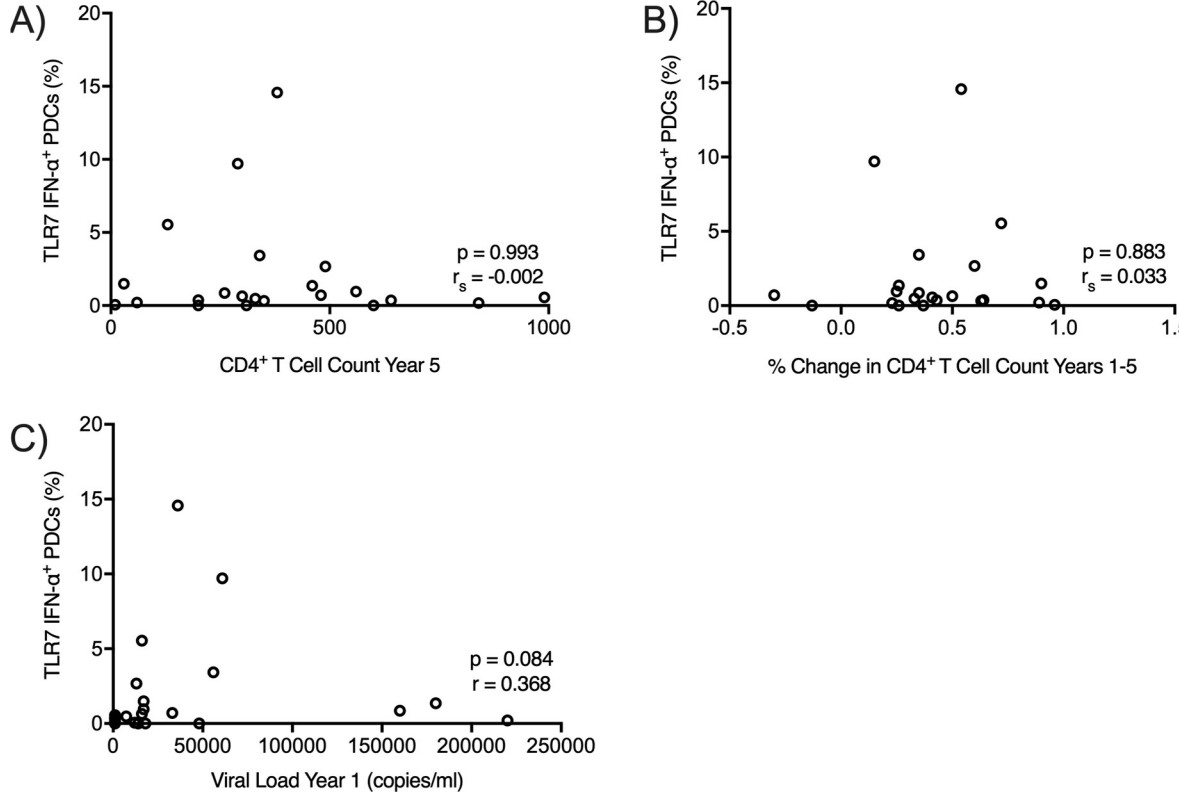

**Fig 6. IFN-α production by TLR7 in PDCs does not correlate with CD4+ decline and viral load in the Amsterdam Cohort.** PDC TLR7 receptors were stimulated by HIV-1ADA. IFN-α production by PDCs was correlated with either the number of CD4+ T cells 5 years post-infection, the percent change in the number of CD4+ cells from 1 to 5 years post infection, or the viral load in year 1. P and $r_s$ values were calculated by the Spearman rank correlation.

## Differences in IFN-α production are not associated with disease progression

We next wished to determine whether greater production of IFN-α by HIV-stimulated PDCs *in vitro* is associated with more rapid disease progression in HIV-infected individuals. Specifically, we asked whether the intrinsic ability of PDC to produce IFN-α prior to infection could be predictive of later disease outcome. In prior work, we observed that PDC production of IFN-α *in vitro* is diminished by a patient's viremia *in vivo*. Therefore, we could not use cells from viremic patients to determine the association between IFN-α production and progressive disease. Similarly, we could not use pre-infection cells from patients receiving anti-retroviral therapy. Therefore, for this analysis, we used samples from the Amsterdam Cohort Studies. In this cohort, cells were collected prior to HIV infection, and patients were followed for many years prior to the availability of ART.

We did not observe a clear association between IFN-α production *in vitro* and disease progression in this cohort. A decrease in CD4+ T cells between year 1 and year 5 post-infection was observed in this cohort with average CD4+ T cell counts between 10 and 990 cells/ml in year 5 (Table 2). We did not observe a correlation between IFN-α production by PDCs and the decline in CD4+ T cells in this cohort ($r_s$ = 0.03, p = 0.883, (Fig 6A and 6B)). In addition, there was no correlation between IFN-α and viral load ($r_s$ = 0.37, p = 0.084, (Fig 6C)). Similarly, we did not observe a correlation between TNF-α production and disease progression (S3A, S3B and S3C Fig).

## Discussion

In the present study, we explored the potential mechanisms of variation in IFN-α production by PDCs in response to TLR7/9 stimulation and the impact of these variations on IFN-α expression in HIV-infected patients. Following TLR7 stimulation, we observed a very strong direct correlation between IFN-α and TNF-α expression by PDCs from all four cohorts. Because the signaling cascades that produce these cytokines bifurcate below MyD88, these data suggest that variations in IFN-α production are not mediated by events downstream of TLR7-MyD88 signaling including IFN-α transcription. Using an RNA40-DOTAP fusion to stimulate PDCs and bypass CD4-mediated entry and virus uncoating, a tight correlation between production of IFN-α and TNF-α in the LTNP/EC and progressor cohorts remained. Taken together, these data are consistent with variations in IFN-α expression originating from within the TLR7-MyD88 receptor complex.

There are numerous reports of associations between IFN-α production by PDCs and viral disease outcomes [19–22]. The source of these variations has in some cases been further associated with polymorphisms within the genes encoding molecules involved in innate sensing. Polymorphisms within the TLR7 or IRF7 genes have been associated with disease outcomes in Hepatitis C, Influenza, and HIV [19–22]. Species-specific differences in single amino acids within IRF7 were thought to be responsible for the lack of progression to AIDS during SIV infection of sooty mangabeys [16], although this observation was not supported by subsequent reports [23]. Our findings are in large part not in agreement with some prior reports suggesting that variations in IFN-α production in human PDCs in response to lentiviral RNA may be related to events within the IRF7 signaling cascade [20]. This discrepancy may be explained by the absence of HIV-uninfected donors with known polymorphisms in exon 3 of the TLR7 gene. Our results suggest that other factors regulate the expression of IFN-α at the level of TLR7. It remains possible that IFN-α is being regulated at the level of the receptor by a number of counter-regulatory, negative feedback mechanisms. Although several of these mechanisms have been described [24, 25], the molecular interactions that result in regulation of this response remain poorly understood. These include the molecular structure of the TLR receptors themselves, the adaptor complex, and other recently identified receptor associated proteins.

We also did not observe a clear association between PDC IFN-α production and disease progression. IFN-α is clearly elevated in vivo during lentiviral infection and may have both a beneficial antiviral effect [26, 27] and detrimental effects in its ability to activate HIV-specific and non-specific immune cells thought to play a role in disease progression [1]. In SIV infection of non-human primates, PDC production of IFN-α has also been linked to immune system activation and progression of disease. Following SIV infection with SIV$_{SM}$ or SIV$_{AGM}$ in sooty mangabeys or African green monkeys, there is an acute type 1 IFN response that is later resolved or dampened despite persistent viremia [23]. This is associated with a non-pathogenic infection in most cases. In contrast, SIV infection of rhesus macaques is associated with persistent production of type I IFNs and disease progression. In humans and macaques, interpreting the cause and effect relationship between disease progression and type I IFN production in vitro is complicated by induction of IFN-α by HIV-1 viremia, the negative impact of HIV-1 viremia on in vitro production of IFN-α based upon feedback inhibition, and in some cases the depletion of PDCs with progressive disease. In addition, many PDCs producing IFN-α may be localized to tissues and are not accessible in the peripheral blood.

Although we did not observe clear evidence of a role for differences in IRF7 signaling in variations in IFN-α production by PDCs nor a clear association of variations in IFN-α production with disease progression, this should not be interpreted as strong evidence against a role

for type I IFNs in lentiviral pathogenesis. These cause and effect relationships are complex and may be best studied through administration of IFN-α during lentiviral infection or blocking its effects. In one study, early blockade of type I IFN signaling accelerated disease progression during SIV infection of rhesus macaques. Administration of IFN-α2 had beneficial effects early during infection, but continued treatment ultimately led to accelerated CD4$^+$ T cell loss, underscoring the complex relationship between type I IFN and disease progression. It remains possible that treatment with other IFN-α subtypes may have a more beneficial role during chronic infection [28].

## Supporting information

**S1 Fig. Gating strategy for stimulated PDCs from all four cohorts.** Total PBMCs were gated by Forward Scatter Area (FSC-A) vs. Side Scatter Area (SSC-A). Dead cells were excluded by Live/Dead fixable violet dead cell stain vs. SSC-A. PDC populations are lineage$^-$, HLA-DR$^+$, and CD123$^+$. The production of IFN-α and TNF-α in PDCs was measured following an 8 hour stimulation.
(TIF)

**S2 Fig. IFN-α production by TLR7 did not correlate with IFN-α production by TLR9 in PDCs.** PDC TLR7 receptors were stimulated by HIV-1$_{ADA}$, and PDC TLR9 receptors were stimulated by CpG A. Cytokine production was inhibited after 4 hours. An anti-IFN-α antibody was used to measure intracellular IFN-α production by flow cytometry. P and r$_s$ values were calculated by the Spearman rank correlation.
(TIF)

**S3 Fig. TNF-α production by TLR7 did not correlate with CD4$^+$ decline nor viral load in the Amsterdam Cohort.** PDC TLR7 receptors were stimulated by HIV-1$_{ADA}$, and cytokine production was inhibited after 4 hours. An anti-TNF-α antibody was used to measure intracellular TNF-α production by flow cytometry. TNF-α production by PDCs was correlated with either the number of CD4$^+$ T cells 5 years post-infection, the percent change in the number of CD4$^+$ cells from 1 to 5 years post infection, or the viral load in year 1. P and r$_s$ values were calculated by the Spearman rank correlation.
(TIF)

## Author Contributions

**Conceptualization:** Kiki Tesselaar, Frank Miedema, Jeffrey Lifson, Mark Connors.

**Data curation:** Andy A. Patamawenu, Nathaniel E. Wright, Tulley Shofner, Sean Evans, Maura M. Manion, Nicole Doria-Rose, Bennett Peterson, Julia Rood, Amy Berkley.

**Formal analysis:** Nathaniel E. Wright, Daniel Mendoza, Julia Rood, Amy Berkley, C. Jason Liang.

**Funding acquisition:** Mark Connors.

**Investigation:** Andy A. Patamawenu, Mark Connors.

**Methodology:** Stephen A. Migueles, Mark Connors.

**Project administration:** Nancy A. Cogliano, Kiki Tesselaar.

**Resources:** Christopher Wilhelm, Julian Bess, Jr., Jeffrey Lifson, Mark Connors.

**Supervision:** Stephen A. Migueles, Mark Connors.

**Writing – original draft:** Mark Connors.

**Writing – review & editing:** Nathaniel E. Wright, Tulley Shofner, Sean Evans, Maura M. Manion, Nicole Doria-Rose, Stephen A. Migueles, Daniel Mendoza, Bennett Peterson, Julia Rood, Amy Berkley, Nancy A. Cogliano, C. Jason Liang, Kiki Tesselaar, Frank Miedema, Jeffrey Lifson, Mark Connors.

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
