## [Decision Letter · Decision Letter 0]

20 Sep 2019

PONE-D-19-21147

Toll-like receptor 7-adapter complex modulates interferon-� production in HIV-stimulated plasmacytoid dendritic cells

PLOS ONE

Dear Dr. Connors,

Thank you for submitting your manuscript to PLOS ONE. After careful consideration by one reviewer and myself, we feel that it has merit but does not fully meet PLOS ONE’s publication criteria as it currently stands. Therefore, we invite you to submit a revised version of the manuscript that addresses the points raised during the review process.

Please address all the comments/suggestions from reviewer 1.

We would appreciate receiving your revised manuscript by Nov 04 2019 11:59PM. To enhance the reproducibility of your results, we recommend that if applicable you deposit your laboratory protocols in protocols.io, where a protocol can be assigned its own identifier (DOI) such that it can be cited independently in the future. For instructions see: http://journals.plos.org/plosone/s/submission-guidelines#loc-laboratory-protocols

We look forward to receiving your revised manuscript.

Kind regards,

Cristian Apetrei, MD, PhD

Academic Editor

PLOS ONE

Journal Requirements:

2. Please include in your Methods section the date ranges over which you recruited participants to this study.

Reviewers' comments:

Reviewer's Responses to Questions

**Comments to the Author**

1. Is the manuscript technically sound, and do the data support the conclusions?

Reviewer #1: Yes

2. Has the statistical analysis been performed appropriately and rigorously? 

Reviewer #1: Yes

3. Have the authors made all data underlying the findings in their manuscript fully available?

Reviewer #1: Yes

4. Is the manuscript presented in an intelligible fashion and written in standard English?

Reviewer #1: Yes

5. Review Comments to the Author

Reviewer #1: In the manuscript the authors investigate the mechanism responsible for a wide variation in the production of INF-a by plasmocytoid dendritic cells (pDCs) in HIV infection. They use four different cohorts: uninfected donors, subjects from the Amsterdam cohort study prior infection, elite controllers that maintain a plasma viral load below 50copies/ml in absence of ART, and progressor subjects. In the study, the authors suggest that the variation in the INF-a production is not due to viral entry efficiency or to the INF-a expression pathway mediated by TLR7/MyD88 complex interaction. Also, they did not observe correlation between the level of INF-a production before infection and disease progression.

The paper is well written and easy to read and understand.

Comments:

1. Table 1: Is the plasma viral load for the progressor subjects below 50 copies/ml?

2. For clarity, the authors should name the panels in the figures and refer to them when describing the results.

3. Figure legends should include more details about the experiments/results

4. Why did the authors use inactivated HIV particles to stimulate pDCs?

5. Did the authors try to stimulate pDC with different amounts of inactivated HIV particles or RNA41 to avoid saturation of CD4 binding or in the expression pathway or INF-a?

6. PLOS authors have the option to publish the peer review history of their article (what does this mean?). If published, this will include your full peer review and any attached files.

Reviewer #1: No

---

## [Author Response · Author response to Decision Letter 0]

4 Nov 2019

1. Table 1: Is the plasma viral load for the progressor subjects below 50 copies/ml?

We have now clarified that the progressor participants had a history of uncontrolled viremia (to distinguish them from LTNP/EC) but were stably suppressed to <50 copies/ml at the time of measurement in the Methods and legend to Table 1.

2. For clarity, the authors should name the panels in the figures and refer to them when describing the results.

The panels in figures with multiple graphs have been labeled and additional details were added to the results section where figures are referenced.

3. Figure legends should include more details about the experiments/results

The figure legends were revised to include a result in the title and methods in the descriptions. 

4. Why did the authors use inactivated HIV particles to stimulate pDCs?

2-aldithriol-inactivated (AT-2) HIV particles have been compared to infectious particles in one study (Sabado et al., Plos One, 2019). They found that inactivated particles are slightly better for antigen presentation. Although the mechanism for this difference is unclear, we primarily used them because we with to avoid a perception of diminished pDC function possibly caused by HIV infection and their use is widespread in this field permitting direct comparisons with our results.

5. Did the authors try to stimulate pDC with different amounts of inactivated HIV particles or RNA41 to avoid saturation of CD4 binding or in the expression pathway or INF-a?

The amount of AT-2 inactivated HIV-1 particles (HIV-1ADA) was titrated in preliminary work (Tilton et al., ref 15). Those titrations indicated that amounts higher than 250 ng of AT-2 inactivated HIV-1 particles did not result in greater amounts of INF-a production. Saturation would be expected to cause less of an increase in IFN-a production for each increase in particle dose, lessening inter-patient variation. However, these variations occurred despite these high doses. Use of a high amount in the present paper permits us to explore the potential mechanism regarding these variations.

---

## [Editor Report · Decision Letter 1]

13 Nov 2019

Toll-like receptor 7-adapter complex modulates interferon-� production in HIV-stimulated plasmacytoid dendritic cells

PONE-D-19-21147R1

Dear Dr. Connors,

We are pleased to inform you that your manuscript has been judged scientifically suitable for publication and will be formally accepted for publication once it complies with all outstanding technical requirements.

With kind regards,

Cristian Apetrei, MD, PhD

Academic Editor

PLOS ONE
---

## [Editor Report · Acceptance letter]

2 Dec 2019

PONE-D-19-21147R1 

Toll-like receptor 7-adapter complex modulates interferon-α production in HIV-stimulated plasmacytoid dendritic cells 

Dear Dr. Connors:

I am pleased to inform you that your manuscript has been deemed suitable for publication in PLOS ONE. Congratulations! Your manuscript is now with our production department. 

With kind regards,

on behalf of

Dr. Cristian Apetrei 

Academic Editor

PLOS ONE